# Role of ^11^C-Methionine PET/CT in ^99m^Tc-Sestamibi-Negative Parathyroid Adenoma: A Case Report

**DOI:** 10.3390/diagnostics11050831

**Published:** 2021-05-04

**Authors:** Jang Yoo, Miju Cheon

**Affiliations:** Department of Nuclear medicine, VHS Medical Center, Seoul 05368, Korea; diva1813@naver.com

**Keywords:** primary hyperparathyroidism, ultrasonography, ^99m^Tc-sestamibi SPECT/CT, ^11^C-methionine PET/CT

## Abstract

We report a case of 16-year-old female primary hyperparathyroidism patient who underwent cervical ultrasonography and ^99m^Tc-sestamibi single photon emission computed tomography/computed tomography, both of which were negative for parathyroid adenoma. Subsequent ^11^C-methionine positron emission tomography/CT showed positive focal uptake suggesting parathyroid adenoma, which then was confirmed pathologically.

Primary hyperparathyroidism (pHPT) is a common endocrine disorder characterized by elevated parathyroid hormone and serum calcium level caused by one or more parathyroid adenomas, parathyroid hyperplasia, or in rare cases, by parathyroid carcinoma [1] (Figure 1). Surgical resection of parathyroid adenoma or hyperplastic parathyroid glands is the curative treatment for pHPT patients. Various imaging modalities are required to avoid extensive surgery and establish a more targeted surgical approach. Preoperative imaging modalities include cervical ultrasonography (US); ^99m^Tc-sestamibi scintigraphy, which has been recently combined with single photon emission computed tomography/computed tomography (SPECT/CT) (Figure 2); and ^11^C-methionine positron emission tomography/CT (^11^C-MET PET/CT) (Figure 3) [2,3,4,5,6,7].

False negative findings of ^99m^Tc-sestamibi scintigraphy can occur due to various reasons. The most common is parathyroid ademona of small size, which allows limited spatial resolution of conventional scintigraphic techiniques [8]. Other reasons for false negative findings are lack of oxyphil cells, parathyroid hyperplasia, multiglandular disease, and high expression of P-glycoprotein [9,10]. In our institute, subsequent ^11^C-MET PET/CT is performed in patients with laboratory findings highly suggestive of pHPT and negative or inconclusive results of US and ^99m^Tc-sestamibi SPECT/CT because the higher resolution of PET/CT could improve localization of small lesions.^11^C-MET is a PET radiopharmaceutical agent that is trapped in the hyperfunctioning parathyroid gland during synthesis of the PTH precursor [11]. The major limitation of this agent is the short physical half-life of ^11^C (20.3 min). To overcome this limitation, an on-site cyclotron is necessary, an instrument that is not available in most clinics. 

## Figures and Tables

**Figure 1 diagnostics-11-00831-f001:**
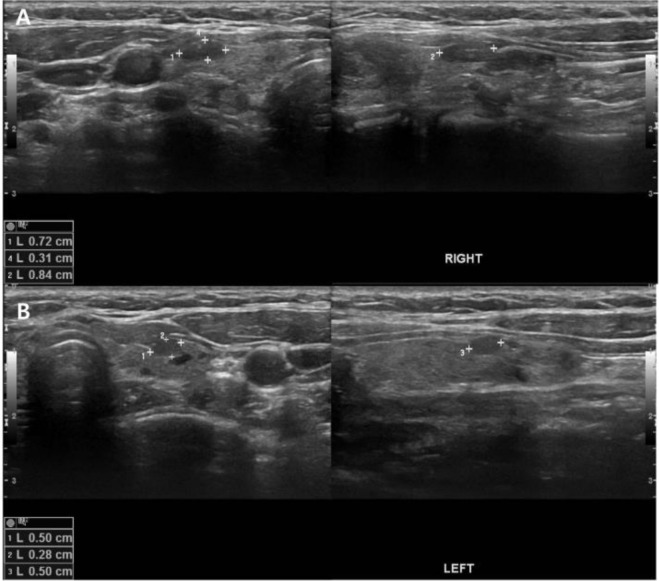
A 16-year-old female patient with elevated parathyroid hormone and serum calcium levels (89.6 pg/mL and 12.8 mg/dL, respectively) underwent initial neck ultrasonography (US) ((**A**) right; (**B**) left) for parathyroid gland evaluation. The images showed heterogeneous echotexture thyroid parenchyme and multiple hypoechoic nodules in both glands but no demonstrable evidence of parathyroid adenoma.

**Figure 2 diagnostics-11-00831-f002:**
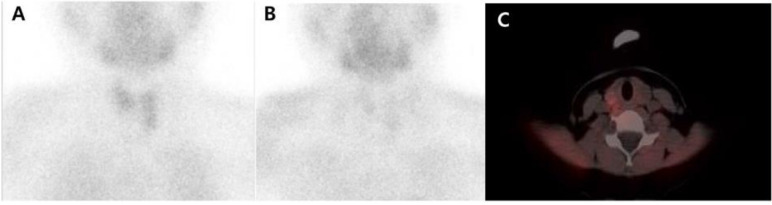
Subsequently, a ^99m^Tc-sestamibi scan was performed at 20 min (**A**) and 150 min (**B**) after radiopharmaceutical injection. There was no abnormal remaining radiotracer uptake lesion suggestive of parathyroid adenoma. Subsequent single photon emission computed tomography/computed tomography (SPECT/CT) (**C**) did not provide additional evidence of parathyroid adenoma.

**Figure 3 diagnostics-11-00831-f003:**
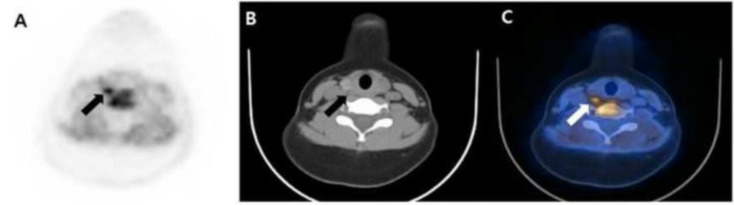
^11^C-Methionine positron emission tomography/CT (^11^C-MET PET/CT) (**A**) PET image; (**B**) CT image; (**C**) fusion PET/CT image) was performed for primary site localization of primary hyperparathyroidism (pHPT) and showed a small lesion in the posterior aspect of the right thyroid gland (arrow). Right parathyroidectomy was performed and pathologically indicated a 0.8-cm-sized parathyroid adenoma. Since the operation, both parathyroid hormone and serum calcium levels have normalized (31.4 pg/mL and 9.8 mg/dL, respectively).

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
