# Peer review of "Role of 11C-Methionine PET/CT in 99mTc-Sestamibi-Negative Parathyroid Adenoma: A Case Report"

_diagnostics, 2021, doi:10.3390/diagnostics11050831_

Round 1
Reviewer 1 Report
The interesting image presented by Jang Yoo and Miju Cheon shows the added value of MET-PET in a case of inconclusive first-line imaging in a patient affected by primary hyperparathyroidism.
While the images are nicely presented, the paper lacks in novelty. The use of PET imaging as a second line approach after inconclusive first-line imaging in primary hyperparathyroidism has been widely described in literature. In view of the relatively high number of retrospective studies, the available meta-analyses and the very recently published EANM procedural guidelines (https://doi.org/10.1007/s00259-021-05334-y) it is not abundantly clear how the case report presented in this paper might provide novel insights on the topic.
Moreover, the clinical description and the discussion are insufficient.
Author Response
We greatly appreiciate the review of our case report and the helpful suggestions. Please find below out point-by-point response to the reviewers’ and editor’s comments, and a decision of the changes made to the manuscript.
Responses to Reviewer #1
The interesting image presented by Jang Yoo and Miju Cheon shows the added value of MET-PET in a case of inconclusive first-line imaging in a patient affected by primary hyperparathyroidism.
While the images are nicely presented, the paper lacks in novelty. The use of PET imaging as a second line approach after inconclusive first-line imaging in primary hyperparathyroidism has been widely described in literature. In view of the relatively high number of retrospective studies, the available meta-analyses and the very recently published EANM procedural guidelines (https://doi.org/10.1007/s00259-021-05334-y) it is not abundantly clear how the case report presented in this paper might provide novel insights on the topic.
- We are fully aware that many review articles about clinical efficacy of 11C-methionine PET/CT have been published recently. This case, which we report currently, is the first experience in our clinical institute, and has greatly helped in the diagnosis of diease and the management of patient.
Moreover, the clinical description and the discussion are insufficient.
- We understand your concern. As you pointed out, we have added the clinical description and the discussion in more detail.
Thank you for your helpful comments.
Reviewer 2 Report
Dear authors,
I think that this case report is interesting and the topic is surely of clinical interest.
However, in the paper there are some points to improve to let the paper more clear and complete.
After these improvement and corrections, I think that the article could be accepted.
COMMENTS
- Title
The title does not fit a case report, try changing it to: "Role of 11C-methionine PET / CT in 99mTc-sestamibi negative parathyroid adenoma: a case report" or something similar. The "incremental role" wording is more suited to an original study or a literature review.
- Figure 1
Please divide the image into two figures: Eco and NM. The parathyroid scintigraphy images are too small and the saturation scale in the spect/ct is too low, making the, difficult to see.
- Figure 2 description
Please add some lines about the alternative to the c11-methionine, such as SPECT image-subtraction after Tc99m-thyroid SPECT when the parathyroid is visible in the SPECT, Fluorocholine, etc.
Furthermore, please add some discussion that support the choice of the c11-methionine.
Author Response
We greatly appreciate the review of our case report and the helpful suggestions. Please find below out point-by-point response to the reviewers’ and editor’s comments, and a decision of the changes made to the manuscript.
Responses to Reviewer #2
Dear authors,
I think that this case report is interesting and the topic is surely of clinical interest.
However, in the paper there are some points to improve to let the paper more clear and complete.
After these improvement and corrections, I think that the article could be accepted.
COMMENTS
Title
The title does not fit a case report, try changing it to: "Role of 11C-methionine PET / CT in 99mTc-sestamibi negative parathyroid adenoma: a case report" or something similar. The "incremental role" wording is more suited to an original study or a literature review.
- As you suggested, we revised the title according to the case report.
Figure 1
Please divide the image into two figures: Eco and NM. The parathyroid scintigraphy images are too small and the saturation scale in the spect/ct is too low, making the, difficult to see.
- As you pointed out, Figure 1 has been newly modified as two figures (Figure 1 and Figure 2).
Figure 2 description
Please add some lines about the alternative to the c11-methionine, such as SPECT image-subtraction after Tc99m-thyroid SPECT when the parathyroid is visible in the SPECT, Fluorocholine, etc.
- We understand your suggestion. We know that 99mTc-pertechnetate is used in the dual-tracer method as it is taken up by functioning thyroid cells and 99mTc-pertechnetate image is subtracted from the 99mTc-sestamibi image revealing hyperfunctioning parathyroid glands, if the thyroid study is performed before or after 99mTc-sestamibi image. However, in our hospital, when parathyroid adenoma is suspected, thyroid scan is not routinely performed. We have added this to the discussion.
Likewise, 18F-fluorocholine can also accumulate in hyperfunctioning parathyroid glands as a cellular proliferation marker, but its use is limited due to cost-effectiveness and local licensing issues. In the future, if we examine these images using various tracers, we will submit to the case report reflecting your helpful opinion.
Furthermore, please add some discussion that support the choice of the c11-methionine.
- As you suggested, we have added some discussion in more detail.
Thank you for your helpful comments.
Reviewer 3 Report
The authors present interesting case of parathyroid adenoma localized by C-11 methionine PET/CT.
Please add a comment on the limitation of availability of C-11 methionine PET in the clinics.
Was enhanced neck CT performed? The findings were?
I did not understand where the references were cited.
Author Response
We greatly appreciate the review of our case report and the helpful suggestions. Please find below out point-by-point response to the reviewers’ and editor’s comments, and a decision of the changes made to the manuscript.
Responses to Reviewer #3
The authors present interesting case of parathyroid adenoma localized by C-11 methionine PET/CT.
Please add a comment on the limitation of availability of C-11 methionine PET in the clinics.
- We understand your concern. We have described the limitation of availability of C-11 methionine PET in the clinics in more detail.
Was enhanced neck CT performed? The findings were?
- The enhanced neck CT was not conducted on this case.
I did not understand where the references were cited.
- As you pointed out, we have revised which references were cited in manuscript.
Thank you for your helpful comments.
Round 2
Reviewer 1 Report
I appreciated the work made by the authors to improve the manuscript.
However, due to the lack of novelty, my general impression is not changed.